# Pharmacological Profile of MP-101, a Novel Non-racemic Mixture of R- and S-dimiracetam with Increased Potency in Rat Models of Cognition, Depression and Neuropathic Pain

**DOI:** 10.3390/cells11244027

**Published:** 2022-12-13

**Authors:** Tiziana Bonifacino, Laura Micheli, Carola Torazza, Carla Ghelardini, Carlo Farina, Giambattista Bonanno, Marco Milanese, Lorenzo Di Cesare Mannelli, Michael W. Scherz

**Affiliations:** 1Department of Pharmacy (DIFAR), Pharmacology and Toxicology Unit, University of Genoa, 16148 Genoa, Italy; 2Inter-University Center for the Promotion of the 3Rs Principles in Teaching & Research (Centro 3R), 56122 Pisa, Italy; 3Department of Neuroscience, Psychology, Drug Research and Child Health, NEUROFARBA-Pharmacology and Toxicology Section, University of Florence, 50139 Florence, Italy; 4Metys Pharmaceuticals c/o Novaremed AG, 4051 Basel, Switzerland; 5IRCCS, Ospedale Policlinico San Martino, 16132 Genoa, Italy

**Keywords:** oxaliplatin-induced neuropathic pain, synaptosomes, allodynia, hyperalgesia, NMDA-induced glutamate release, allosteric modulation

## Abstract

The racemic mixture dimiracetam negatively modulates NMDA-induced glutamate release in rat spinal cord synaptosomal preparations and is orally effective in models of neuropathic pain. In this study, we compared the effects of dimiracetam, its R- or S-enantiomers, and the R:S 3:1 non-racemic mixture (MP-101). In vitro, dimiracetam was more potent than its R- or S-enantiomers in reducing the NMDA-induced [^3^H]D-aspartate release in rat spinal cord synaptosomes. Similarly, acute oral administration of dimiracetam was more effective than a single enantiomer in the sodium monoiodoacetate (MIA) paradigm of painful osteoarthritis. Then, we compared the in vitro effects of a broad range of non-racemic enantiomeric mixtures on the NMDA-induced [^3^H]D-aspartate release. Dimiracetam was a more potent blocker than each isolated enantiomer but the R:S 3:1 non-racemic mixture (MP-101) was even more potent than dimiracetam, with an IC_50_ in the picomolar range. In the chronic oxaliplatin-induced neuropathic pain model, MP-101 showed a significantly improved anti-neuropathic profile, and its effect continued one week after treatment suspension. MP-101 also performed better than dimiracetam in animal models of cognition and depression. Based on the benign safety and tolerability profile previously observed with racemic dimiracetam, MP-101 appears to be a novel, promising clinical candidate for the prevention and treatment of several neuropathic and neurological disorders.

## 1. Introduction

Dimiracetam, a racemic, bicyclic rigid analog of piracetam, previously showed cognition-enhancing properties in preclinical in vivo experiments [1]. At higher single oral doses, dimiracetam exhibited anti-depressant effects in the forced swimming test [2] and anti-hyperalgesic and anti-allodynic broad-spectrum activity in various models of neuropathic pain induced by antiretroviral or chemotherapeutic drugs and in the mono-iodoacetate (MIA)-induced osteoarthritis model [3]. 

Dimiracetam belongs to the so-called racetam family of nootropic drugs, whose pharmacology was explored in many cognitive and neurological disorders. Piracetam, oxiracetam, and aniracetam are commercialized members of this family that display promising pro-cognitive effects in animals but fail to provide robust evidence of improving cognition in degenerative dementia. While their clinical use remains limited [4], they are widely used as nutritional supplements [5]. Levetiracetam [6] is a widely-used antiepileptic drug [6] and there is controversial evidence of its potential benefit with headaches [7,8,9] and other forms of neuropathic pain [10,11]. A report on the efficacy of nefiracetam in tactile and thermal hyperalgesia in mice [12] suggests that chemical modifications of the racetam structure can confer robust anti-neuropathic pain activity.

The molecular mechanism of action of racetams is poorly defined and their pharmacology appears to change significantly with minor chemical modifications [13]. Initially, their structural similarity to a cyclic derivative of GABA suggested possible GABA-mimetic action [14]. However, this mechanism, as well as possible antioxidant and effects [15,16] or acetylcholine receptor density-increasing properties [17] are no longer much discussed. Clearly, though, racetams act centrally to modulate spinal and brain pre- and post-synaptic neurotransmission [3,18,19,20,21]. Levetiracetam and brivaracetam are reported to act via the synaptic vesicle 2A protein (SV2A) [22]; however, levetiracetam appears to exploit additional mechanisms, including the interaction with α-amino-3-hydroxy-5-methyl-4-isoxazolepropionic acid (AMPA) and kainate-gated ion channels [23]. 

We previously showed that dimiracetam counteracted the N-methyl-D-aspartate (NMDA)-induced release of glutamate in spinal cord synaptosomal preparations, possibly by negatively modulating specific subtypes of the NMDA receptor containing the proton-sensitive GluN1 subunit and the GluN2A and GluN2B subunits [3]. We postulated that the reduction of the NMDA-induced glutamate release may have resulted in a de-winding of the excitatory transmission-sustained wind-up, responsible for reinforcing pain perception in most neuropathic conditions [3,24]. 

Drug-induced peripheral neuropathy (DIPN) is commonly caused by widely prescribed chemotherapeutic, antimicrobial, and anti-viral drugs [25]. Current first-line DIPN treatments include amitriptyline, duloxetine, pregabalin, and gabapentin; second-line treatments include lidocaine plasters, capsaicin patches or tramadol [26]. None of these treatments are satisfactory; therefore, novel and more efficacious agents are urgently sought. In this contest, dimiracetam is a promising developing candidate for neuropathic pain. It is orally active in rat models induced by oxaliplatin, paclitaxel, vincristine, sorafenib [27,28,29], 2′-3′-dideoxycytidine (ddC) [3], or trauma [29], or in a rat model of articular pain [3]. 

In the present experiments, we further investigated the anti-glutamate properties of dimiracetam and its isolated R- and S-enantiomers. Surprisingly, we found that, while R-dimiracetam was more potent than the S-enantiomer, both enantiomers were less potent than racemic dimiracetam by five- and 15-fold, respectively. To better understand the reasons for this unorthodox finding, we decided to explore *in vitro* and *in vivo* the properties of several non-racemic mixtures of the R- and S-dimiracetam enantiomers, eventually culminating in the identification of the R:S 3:1 mixture (MP-101) as a novel development candidate, endowed with higher potency than racemic dimiracetam, for prevention of oxaliplatin-induced symptoms of peripheral neuropathy. 

## 2. Materials and Methods

### 2.1. Animals

Male Sprague-Dawley rats or male CD-1 mice (Harlan, Varese, Italy) weighing 150–200 g and 20–25 g, respectively at the beginning of the experimental procedure were used. Animals were housed in the animal houses of the University of Florence (Ce.SAL) and University of Genoa (Dept. of Pharmacy—DIFAR) and used at least 1 week after their arrival. Four rats or ten mice were housed per cage (size 26 cm × 41 cm). Animals were fed with a standard laboratory diet and tap water ad libitum and kept at 23 ± 1 °C with a 12 h light/dark cycle (light at 7 a.m.). Experiments involving animals were carried out according to the ARRIVE guidelines [30] and to what was declared in “Institutional Review Board Statement”. All efforts were made to minimize animal suffering and reduce the number of animals used. All the *in vivo* assessments were made by researchers blinded to animal treatments.

### 2.2. Materials

Racemic dimiracetam (R:S 1:1), R-enantiomer and S-enantiomer of dimiracetam used in *in vitro* and *in vivo* experiments were supplied by Metys Pharmaceuticals AG, Basel Switzerland. The two enantiomers had been prepared from R- or S-pyroglutamic acid, respectively, as reported previously [29].

The following different non-racemic mixtures, with R-dimiracetam enriched with respect to S-dimiracetam, (R:S), were used: 2:1, 2.5:1, 3:1 (MP101), 3.25:1, 3.5:1, 4:1, 5.7:1, 9:1, and 19:1.

In all *in vivo* experiments, drugs were dissolved in 1% carboxymethylcellulose sodium salt (CMC; Sigma-Aldrich, Milano, Italy) and administered orally at 10 mL kg^−1^ body weight. Control rats received an equal volume of CMC.

### 2.3. Synaptosome Purification

Adult male Sprague Dawley rats were euthanized, and the spinal cord was rapidly removed. Isolated axon terminals (synaptosomes) were purified as previously reported [31]. The tissue was homogenized in 10 vol. of 0.32 M sucrose, buffered at pH 7.4 with Tris-HCl, using a glass-Teflon tissue grinder (clearance 0.25 mm). The homogenate was centrifuged (5 min, 1000× *g* at 4 °C) to remove nuclei and debris, and the supernatant was gently layered on a discontinuous Percoll (Sigma-Aldrich, St. Louis, MO, USA) gradient (2%, 6%, 10%, and 20% *v*/*v* in Tris-buffered 0.32 M sucrose). After centrifugation at 33,500× *g* for 5 min, the layer between 10 and 20% Percoll^®^ (synaptosomal fraction) was collected and washed in a physiological medium with the following compositions (mM): NaCl, 140; KCl, 3; MgSO_4_, 1.2; NaH_2_PO_4_, 1.2; NaHCO_3_, 5; CaCl_2_, 1.2; 4-(2-hydroxyethyl)-1-piperazineethanesulfonic acid (HEPES), 10; glucose, 10; pH 7.4 and centrifuged (20,000× *g* for 15 min). Synaptosomes were resuspended in the physiological medium for release experiments. 

### 2.4. Release Experiments

Synaptosomes were incubated at 37 °C for 15 min in the presence of 0.1 µM [^3^H]D-aspartate ([^3^H]D-Asp; 0.5 Ci/mmol; Perkin Elmer Boston, MA, USA), a non-metabolizable analog of glutamate widely utilized to label the endogenous glutamate pools of nerve terminals [32]. 

Aliquots of the synaptosomal suspension were distributed on microporous filters placed at the bottom of a set of 24 parallel superfusion units maintained at 37°C (Superfusion System, Ugo Basile, Comerio, Varese, Italy), and up-down superfused with the physiological medium at 0.5 mL/min [33,34]. Superfusion was continued for 48 min. After 37 min of superfusion to equilibrate the system, 11 one-minute samples were collected. A physiological medium Mg^2+^-free and supplemented with 10 µM NMDA and 1 µM glycine (reported as NMDA) was added at t = 38 min after collecting the first sample. When appropriate, dimiracetam, R-dimiracetam, S-dimiracetam, and the non-racemic R:S mixtures of the two enantiomers were added concomitantly to NMDA. Appropriate controls were always run in parallel. 

At the end of the experiment, radioactivity was determined in each sample collected and in synaptosomes on the superfused filters by liquid scintillation (Ultima Gold, Perkin Elmer, Milan, Italy) counting. Tritium released in each sample was calculated as fractional rate × 100 (percentage of the total synaptosomal neurotransmitter content at the beginning of the respective sample collection). Drug effects were evaluated by calculating the ratio between the efflux in the ninth or tenth sample (where the maximum effect of NMDA was generally reached) and that of the first sample. This ratio was compared to the corresponding ratio obtained under control conditions. 

### 2.5. Sodium Monoiodoacetate-Induced Osteoarthritis Model 

Unilateral osteoarthritis was induced by injection of sodium monoiodoacetate (MIA; Sigma-Aldrich, Milan, Italy) into the tibiotarsal joint [35]. On day 1, rats were lightly anaesthetized by 2% isoflurane, the left leg skin was sterilized with 75% ethyl alcohol and the lateral malleolus located by palpation; then, a 28-gauge needle was inserted vertically to penetrate the skin and turned distally for insertion into the articular cavity at the gap between the tibiofibular and tarsal bone until a distinct loss of resistance was felt. Two milligrams of MIA in 25 μL saline were delivered into the left articular cavity. Control rats were treated with an equal volume of saline. Racemic dimiracetam (R:S 1:1), pure enantiomers and R:S 3:1 or R:S 1:3 enantiomeric mixtures were acutely administered by oral route at the dose of 300 mg kg^−1^; pain-like behavior (mechanical hyperalgesia) was assessed on day 16, by when osteoarthritic-like focal lesions in articular cartilage and subchondral bone thickening had been established by MIA [36].

### 2.6. Oxaliplatin-Induced Neuropathic Pain Model

Oxaliplatin (2.4 mg kg^−1^; Carbosynth, Compton, UK) was injected intraperitoneally (i.p.) on days 1–3, 6–10 and 13–15 for a total of 11 injections, according to Cavalletti [37] with minor modifications concerning the days of oxaliplatin administration [38]. Oxaliplatin was dissolved in a 5% glucose solution. Control animals received an equivalent volume of vehicles. Dimiracetam (R:S 1:1) and MP-101 (R:S 3:1 non-racemic mixture) were administered twice daily by oral gavage at the doses of 15 and 50 mg kg^−1^ from day 1 to day 22. Pain-like behavior (mechanical hyperalgesia; mechanical and cold allodynia) was assessed before and 1 h after the morning administration on days 1, 3, 8, 15, 22 and after 1 week of treatment wash-out (day 28).

### 2.7. Assessment of Mechanical Hyperalgesia (Paw Pressure Test)

The nociceptive threshold of rats was determined with an analgesimeter (Ugo Basile, Varese, Italy), as previously described [39]. A constantly increasing pressure was applied to a small area of the dorsal surface of the hind paw using a blunt conical probe by a mechanical device. Mechanical pressure was increased until vocalization or a withdrawal reflex occurred while rats were lightly restrained. Vocalization or withdrawal reflex thresholds were expressed in grams. The cut-off was set at 200 g.

### 2.8. Assessment of Mechanical Allodynia (Von Frey Test)

Mechanical allodynia was measured with an electronic Von Frey hair unit (Ugo Basile, Varese, Italy) as described previously [40]. Briefly, rats were placed in 20 cm × 20 cm Plexiglas boxes equipped with a metallic mesh floor, 20 cm above the bench. Rats were allowed to get used to the environment for 15 min before the test. The withdrawal threshold was evaluated by applying forces ranging from 0 to 50 g with a 0.2 g accuracy. Punctuate stimulus was delivered to the mid-plantar area of the ipsilateral and the contralateral posterior paw from below the mesh floor through a plastic tip and the paw withdrawal threshold was automatically displayed on the screen. The paw sensitivity threshold was defined as the minimum force required to obtain a robust and immediate withdrawal reflex of the paw. Voluntary movements associated with locomotion were not considered a withdrawal response. Stimuli were applied to each anterior paw at 5 s intervals. Measurements were repeated 5 times and the final value was obtained by averaging [41].

### 2.9. Assessment of Thermal Allodynia (Cold Plate Test)

Thermal allodynia was assessed using the cold-plate apparatus (Ugo Basile, Varese, Italy). With minimal animal–handler interaction, rats were taken from home cages and placed onto the cold surface maintained at a constant temperature of 4 ± 1 °C. Ambulation was restricted by a cylindrical Plexiglas chamber (diameter: 10 cm, height: 15 cm), with open top. Pain-related behavior (paw lifting or licking) was observed, and the time (in seconds) of the first sign was recorded. The cut-off time latency was set at 30 s [42].

### 2.10. Assessment of Amnesic-like Behavior in Mice (Passive Avoidance Test)

The effect of racemic dimiracetam and different enantiomeric mixtures of dimiracetam on scopolamine-induced amnesia in mice was assessed by the passive avoidance test, according to the step-through method described by Jarvik and Kopp [43]. The test was performed in an apparatus consisting of a two-compartment acrylic box with a lighted compartment connected to a darkened one by a guillotine door. In the original method, mice received a punishing electrical shock as soon as they entered the dark compartment, while in our modified method, after entry into the dark compartment, mice receive a non-painful punishment consisting of a fall (from 40 cm) into a cold-water bath (10 °C). For this purpose, the dark chamber was constructed with a pitfall floor. On day 1, mice receive the punishment when entering the dark room in the training session and remembered it in the test session on day 2, unless their memory was impaired by the amnesic drug. Mice who did not enter after 60 s latency in the training session were excluded from the experiment; about 20–30% of mice were excluded from each group. Scopolamine (1.5 mg kg^−1^) or saline vehicle was injected intraperitoneally immediately after the punishment on day 1. Single oral doses of the investigated drugs ((3, 10 and 30 mg kg^−1^) dissolved in 1% CMC) or vehicle were administered by gavage 30 min before the training session on day 1. The maximum entry latency allowed in the retention session was 180 s.

### 2.11. Assessment of Depressive-like Behavior in Mice (Forced Swimming Test)

The antidepressant-like effects of dimiracetam and its non-racemic enantiomeric mixtures were assessed in mice by the forced swimming test, as described by Porsolt [44]. Briefly, mice were dropped individually into glass cylinders (height: 25 cm, diameter: 10 cm) containing 6 cm of water maintained at 22–23 °C and left there for 6 min. A mouse was judged to be immobile when it floated in the water, in an upright position, and made only small movements to keep its head above water. The duration of immobility was recorded during the last 4 min of the 6 min test. A decrease in the duration of immobility was taken as a measure of an antidepressant-like effect. Single oral doses of the investigated drugs (3, 10 and 30 mg kg^−1^) dissolved in CMC or vehicle were administered by gavage 25 min before the test was performed.

### 2.12. Statistical Analysis

Data are expressed as mean ± standard error of the mean (S.E.M.). Two-tailed Student’s t-test or ANOVA followed by the Bonferroni post-hoc test was applied when appropriate. *p*-values < 0.05 were considered significant. Statistical analysis was carried out using GraphPad Prism (GraphPad Software, La Jolla, CA, USA) or OriginPro 9 software (OriginLab, Northampton, MA, USA). Curves and graphs were fitted using the Sigma Plot 10, GraphPad Prism, or OriginPro 9 software, as appropriate.

## 3. Results

### 3.1. Effects of Dimiracetam (R:S 1:1) and Its Individual R- and S-enantiomers on the NMDA-Evoked Release of [^3^H]D-Asp from Rat Spinal Cord Synaptosomes

Previous studies showed that NMDA stimulates the release of glutamate from synaptosomes purified from rat spinal cord and that dimiracetam reduces the NMDA-induced glutamate release [3,45,46]. Here, we investigated the effect of the individual R- or S-enantiomers of dimiracetam. 

Racemic dimiracetam and its individual R- and S-enantiomers (0.1 nM–10 µM) concentration-dependently reduced the [^3^H]D-Asp release stimulated by 10 µM NMDA + 1 µM glycine in Mg^2+^-free medium from rat spinal cord synaptosomes (Figure 1a). The IC_50_ value of racemic dimiracetam calculated from the curve fitting was 16.26 ± 2.09 nM. The IC_50_ value of the R-enantiomer of dimiracetam was 87.45 ± 9.96 nM, i.e., about five-fold higher than the value of racemic dimiracetam. S-dimiracetam IC_50_ was 248.50 ± 43.79 nM. In contrast to racemic dimiracetam or the R-enantiomer of dimiracetam, S-dimiracetam was only partially effective as only about 60% of the NMDA-induced [^3^H]D-Asp release was reduced by S-dimiracetam at the highest concentration tested (100 µM).

### 3.2. Effect of a Single Administration of Racemic Dimiracetam (R:S 1:1) and Its Isolated R- and S-enantiomers in an MIA-Induced Osteoarthritis Model

Previous studies showed that single oral doses of racemic dimiracetam (R:S 1:1) reduce pain-like behavior in the rat MIA model [3]. Here, we repeated the assay and by testing separately the individual R- and S-enantiomers. 

Sixteen days after the damage evoked by MIA, the weight tolerated by the animals on the ipsilateral paw was significantly reduced in comparison to the control group (vehicle + vehicle), as shown in Figure 1b. Racemic dimiracetam (R:S 1:1) administered at the dose of 300 mg kg^−1^ significantly improved the weight tolerated by rats on the ipsilateral paw with a peak of efficacy at 15 min after treatment. The effect slightly decreased over time, but it was still significant for up to 60 min (Figure 1b). The R-enantiomer, tested at the same dose of 300 mg kg^−1^, showed a lower response as compared to the anti-hyperalgesic effect evoked by the racemate, and its effect lasted up to 45 min after treatment. The S-enantiomer was poorly active, showing a smaller but still statistically significant effect 15 min after administration (Figure 1b).

### 3.3. Comparison of Several Enantiomeric Mixtures in a Ratio of R- to S- Varying from 2:1 to 19:1, in Inhibiting the NMDA-Evoked Release of [^3^H]D-Asp from Rat Spinal Cord Synaptosomes

The above experiments depicted an unusual situation in which both R- and S-dimiracetam were less potent than the racemate in reducing the release of [^3^H]D-Asp evoked by NMDA and in reducing the hyperalgesic response in MIA-treated rats, suggesting that the simultaneous presence of the two enantiomers is required to improve the potency. This led us to explore further, *in vitro* and *in vivo*, the effect of mixtures with a different ratio of R- to S-enantiomers, with an increasing proportion of the more potent R-enantiomer.

Figure 2a shows the inhibitory effect of increasing concentrations (0.1 nM–100 µM) of the R:S 2:1 mixture on [^3^H]D-Asp release elicited by NMDA. When fitting the points, a biphasic curve emerged, showing both high- and low-affinity branches with a plateau between 10 nM and 1 µM. The high-affinity portion of the curve was characterized by an IC_50_ of 3.02 ± 0.36 nM, while the low-affinity portion by an IC_50_ of 79.40 ± 9.11 µM. Figure 2b shows the effects of the R:S 3:1 mixture. Again, the curve turned biphasic, with a plateau between 1 nM and 1 µM. The curve’s high- and low-affinity portions were distinguished by IC_50_s of 0.21 ± 0.03 nM and 14.89 ± 3.38 µM, respectively. Both the R:S 2:1 and 3:1 mixtures abolished the [^3^H]D-Asp release evoked by NMDA at the highest tested concentration (100 µM). We compared the concentration–response curves of racemic dimiracetam with the R:S 2:1 (Figure 2c) and R:S 3:1 (Figure 2d) enantiomeric mixtures, respectively. The high-affinity portion of the R:S 3:1 mixture lies at the left of the racemic dimiracetam curve, thus indicating a higher potency of R:S 3:1 than the racemate. This was not the case with the R:S 2:1 mixture.

We explored the behavior of an expanded range of enantiomeric mixtures, maintaining an excess of the R-enantiomer and comparing these with R:S 3:1. Figure 3a shows the NMDA-stimulated release of [^3^H]D-Asp in the absence and the presence of 10 nM of R:S mixtures in a ratio of 3:1, 4:1, 5.7:1, 9:1 or 19:1, respectively. As before, the R:S 3:1 mixture almost halved the NMDA effect; the R:S 4:1, 5.7:1, and 9:1 mixtures were inactive at this concentration; the 19:1 mixture, as expected, significantly reduced the [^3^H]D-Asp release evoked by NMDA to an extent similar to that of pure R-dimiracetam. 

To further fine-tune the modulation of NMDA-induced [^3^H]D-Asp release around the R:S 3:1 ratio, we investigated several R:S ratios between 2:1 and 4:1 at the two concentrations of 3 or 10 nM. Figure 3b reports the potentiation of the [^3^H]D-Asp release induced by NMDA in the presence or absence of 3 nM of the R:S 2:1, 2.5:1, 3:1, 3.25:1, 3.5:1 and 4:1 enantiomeric mixtures. The R:S 3.5:1 and 4:1 mixtures did not significantly reduce the NMDA effect, while the other R:S mixtures did, with the R:S 3:1 and the 3.25:1 mixtures displaying greater potency than the other ratios. Figure 3c shows the effects of the same mixtures at 10 nM concentration. The R:S 4:1 mix was again inactive at this concentration, while the other R:S mixtures, including the R:S 3.5:1, inactive at 3 nM, significantly reduced the NMDA potentiation.

### 3.4. Effect of a Single Administration of R:S 3:1 and R:S 1:3 Enantiomeric Mixtures of Dimiracetam in the Rat Model of MIA-Induced Osteoarthritis

To assess whether an excess of the R-enantiomer versus S-enantiomer was a requisite for greater potency, we compared the anti-hyperalgesic effect of the R:S 1:3 and 3:1 mixtures in vivo in the MIA osteoarthritis paradigm. Acute administration of 300 mg kg^−1^ R:S 3:1 was able to significantly increase rats’ pain threshold, with a peak at 15 min after administration and the effect lasting up to 45 min. The R:S 1:3 enantiomeric mixture, at the same dose of 300 mg kg^−1^, showed a lower potency than the R:S 3:1 enantiomeric mixture (Appendix A).

### 3.5. Effect of a Repeated Treatment with R:S 1:1 and R:S 3:1 (MP-101) Enantiomeric Mixtures of Dimiracetam in the Rat Oxaliplatin-Induced Neuropathic Pain Model

Oxaliplatin injected intraperitoneally on days 1–3, 6–10, and 13–15, for a total of 11 injections, significantly reduced the weight tolerated by the rats on a posterior paw starting from day 3 after the beginning of the treatment. The paw-withdrawal threshold in this treatment group decreased over time reaching values of about 45 g at day 15 in comparison with the 65 g recorded in control animals (Figure 4a,c). Twice daily oral treatment with a low dose (50 mg kg^−1^) or a very low dose (15 mg kg^−1^) of either R:S 1:1 (racemate) or R:S 3:1 (MP-101) enantiomeric mixtures, from day 1 up to day 22 (inclusive) significantly increased the paw-withdrawal threshold of oxaliplatin-treated rats compared to the vehicle-treated group at both tested dosages for both mixtures. Slightly greater improvements were noted for the higher of the two doses (Figure 4a,b). Importantly, the effectiveness of R:S 3:1 at either of the two doses was significantly higher in comparison to either the corresponding doses of the racemate, starting from day 13 and up to day 22.

The preventive effects of racemic R:S 1:1 and R:S 3:1 were also evaluated on day 28, i.e., 6 days after the last oral dose. At this time point, oxaliplatin-treated rats still showed a decrease in the weight tolerated on a posterior paw in comparison to vehicle + vehicle-treated rats. Rats treated with 15 mg kg^−1^ b.i.d. R:S 3:1 displayed a paw-withdrawal threshold significantly improved while racemic dimiracetam (R:S 1:1) was ineffective at either doses (Figure 4b). The 50 mg kg^−1^ b.i.d. dose of R:S 3:1 was fully effective against oxaliplatin-induced hypersensitivity compared to racemic R:S 1:1 or vehicle-treated rats 6 days after the last oral administration (Figure 4b). 

The response to a non-noxious mechanical stimulus was also evaluated on days 1, 3, 8, 15 and 22 by the von Frey test (Figure 4c,d). Oxaliplatin significantly decreased the paw-withdrawal threshold of the animals starting from day 15 until day 22. Repeated oral treatments with R:S 1:1 or R:S 3:1 reduced the paw-withdrawal response, counteracting oxaliplatin-induced allodynia, independent of the administered dose (Figure 4c,d). Similarly to the results observed in the paw pressure test, both oral R:S 3:1 doses remained highly effective after one week of treatment washout, as compared with the racemate, which no longer showed effects (Figure 4c,d).

Finally, the response to a thermal non-noxious stimulus was also assessed by the cold plate test (Figure 4e,f). Cold hypersensitivity emerged starting from day 8 when the licking latency recorded in oxaliplatin-treated animals reached values of about 15 s, as compared to 20 s for the control group. On day 8, both R:S 1:1 and R:S 3:1 significantly increased the licking latency (Figure 4e,f). Throughout the subsequent measures performed on days 15 and 22, R:S 3:1 showed a higher potency in comparison to R:S 1:1 tested at the same doses. Once again, the greater effectiveness of both dosages of R:S 3:1 was confirmed also in the experiments performed 6 days after the last oral administration (Figure 4e,f).

The effects of R:S 1:1 and R:S 3:1 on mechanical hyperalgesia and mechanical and thermal allodynia were also confirmed 1 h after the morning administration on days 5, 8, 15 and 22 (Appendix A). Compared to the pre-administration results on the same morning, the superior potency of R:S 3:1 over R:S 1:1 was still apparent, even though a little bit less outstanding than on day 28. 

### 3.6. Effect of a Single Administration of R:S 1:1, 2:1 and 3:1 Mixtures against Scopolamine-Induced Amnesia

Dimiracetam (R:S 1:1) and the two non-racemic R:S 2:1 and R:S 3:1 mixtures were tested for their ability to revert scopolamine-induced amnesia in the mouse passive avoidance test. For each mixture, three different dosages of 3, 10 and 30 mg kg^−1^ were used, as shown in Figure 5. All tested mixtures prevented the scopolamine-induced amnesic effect in a dose-dependent manner; in particular, the non-racemic R:S 3:1 (MP 101) mixture was about three-fold more potent than the racemate, showing a similar effect at 10 and 30 mg kg^−1^, respectively. Moreover, the higher dose of R:S 3:1 fully prevented the scopolamine effect, as the entrance latency of the animals was comparable to that of control animals treated with saline during the retention test (Figure 5).

### 3.7. Effect of Single Administration of R:S 1:1, 2:1 and 3:1 Mixtures in a Depression Mouse Model

Finally, the R:S 1:1, 2:1 and 3:1 mixtures were examined in the forced swimming test at the doses of 10, 30 or 100 mg kg^−1^ to assess their potential antidepressant-like effects. 

All the higher dosages (100 mg kg^−1^) of the three mixtures displayed a potential antidepressant effect, with R:S 3:1 (MP-101) increasing the mouse mobility time up to 100 s, as compared to the averaged 70 s of control animals. While the lowest dose 10 mg kg^−1^ of thethree mixtures was ineffective, a statistically significant effect was obtained with the intermediate dose (30 mg kg^−1^ ) of both R:S 2:1 and 3:1 mixtures (Figure 6). 

## 4. Discussion

Dimiracetam is one of the latest members of the racetam family of nootropic drugs; it is a rigid analog of piracetam and was developed as a racemate to prevent or treat HIV-induced neuropathic pain, demonstrating excellent safety and tolerability in preclinical and Phase 1/2 clinical trials [47].

In previous preclinical *in vivo* studies in rodents [1], dimiracetam displayed cognition-enhancing properties with a potency 10-30-fold higher than oxiracetam [1]. Later studies at higher oral doses established dimiracetam’s anti-depressant actions in the forced swimming test [2,3]. In animal models of painful neuropathy induced by antiretroviral drugs or oxaliplatin and in the MIA-induced osteoarthritis model, single doses of dimiracetam were more efficacious than pregabalin, reverting the painful condition to control animal values, with its broad spectrum of anti-hyperalgesic and anti-allodynic effects being more pronounced after chronic oral administration [3]. 

Previous studies with purified synaptosomes demonstrated the presence and the role of presynaptic release-regulating auto- and hetero-receptors in the CNS [48]. Dimiracetam and structurally related nootropic compounds interact with these presynaptic release-regulating receptors, particularly the ionotropic glutamate receptors, and modulate the release of neurotransmitters from brain and spinal cord synaptosomes [3].

Nootropics such as oxiracetam and aniracetam increased the NMDA-induced release of glutamate, noradrenaline, and acetylcholine, preventing the inhibition due to kynurenic acid [19,20,49]. On the other hand, dimiracetam dose-dependently reduced the NMDA-induced glutamate release in both the hippocampus and spinal cord, with low nanomolar potency in the latter case [3]. 

Here, we investigated the pharmacological effects of racemic dimiracetam and its enantiomers in antagonizing the NMDA-induced [^3^H]D-Asp release from rat spinal cord synaptosomes *in vitro*. We discovered that the simultaneous presence of R- and S-dimiracetam is required to improve effects at the NMDA receptor, with the S-enantiomer being less potent (about three-fold) than R-enantiomer and the R-enantiomer being less potent (approximately five-fold) than racemic dimiracetam (Figure 1a). 

We then investigated the full dose–response curves in rat spinal synaptosomes of the R:S 3:1 and the R:S 2:1 mixtures and observed a biphasic effect with both mixtures, being high- and low-affinity branches and an intermediate plateau were readily apparent. The high-affinity portion of the R:S 3:1 mixture was more potent than racemic dimiracetam curve, whilst the high-affinity portion of the R:S 2:1 mixture was approximately equipotent to racemic dimiracetam. The plateau effect of the two mixtures was undoubted, spanning three to four logarithmic concentration decades. To further characterize the behavior of non-racemic dimiracetam, a broader range of non-racemic mixtures spanning from R:S 2.5:1 to 19:1 was assessed, comparing single fixed concentrations of each mixture. Only the mixtures with an enantiomeric ratio close to R:S 3:1 were more effective than racemic dimiracetam. 

We previously described the efficacy of racemic dimiracetam in rodent models of neuropathy induced by antiretroviral [3] or chemotherapeutic drugs [27,28] or trauma [29]; we also noted that cerebral cortex nerve terminals of oxaliplatin-treated rats exhibit higher baseline and NMDA-induced [^3^H]D-Asp release [50]. In the present work, we compared the R:S 3:1 mixture and racemic dimiracetam in the MIA-induced osteoarthritis and in the oxaliplatin-induced neuropathic pain models in rats. Our *in vivo* results showed very close correspondence to the *in vitro* findings. First, we used the MIA model to confirm that racemic dimiracetam is more potent than each R- or S-enantiomer in preventing mechanical hyperalgesia. In addition, we showed that non-racemic enantiomeric mixtures enriched with R-dimiracetam are required to achieve a higher potency, as our results confirmed that the R:S 1:3 mixture, enriched with S-dimiracetam, is significantly less effective than the R:S 3:1 (MP-101) mixture.

In the rat model of oxaliplatin-induced neuropathic pain, we compared the effects on mechanical hyperalgesia and mechanical or thermal allodynia of a low (50 mg kg^−1^ b.i.d.) or a very low (15 mg kg^−1^ b.i.d.) dose of racemic dimiracetam and the same doses of the R:S 3:1 mixture (MP-101) of dimiracetam enantiomers. MP-101 was clearly more effective than racemic dimiracetam at all time points up to day 22. In particular, the 15 mg kg^−1^ dose of MP-101 is more effective than the 50 mg kg^−1^ dose of racemic dimiracetam. Following a 6-day treatment-free period, the MP-101-treated groups still exhibit statistically significant effects vs. the vehicle treatment arm, whereas the treatment effect is no longer evident in the racemic dimiracetam groups. 

Finally, in mouse models of memory and depression, our *in vivo* results again demonstrated the superiority of MP-101 over racemic dimiracetam. In these experiments, we also tested the R:S 2:1 mixture and found higher therapeutic effects with respect to dimiracetam.

It is rare but not unprecedented to consider non-racemic mixtures of an active drug to improve its properties. For example, a mixture containing a 60–77% enantiomeric excess of the indacrinone R-enantiomer improved its pharmacological profile over its S-enantiomer alone [51]. Additionally, using a 3:1 ratio of levo- and dextrorotatory enantiomers of bupivacaine optimized the equilibrium between the anesthetic and cardiotoxic properties [52,53]. In both examples, though, the two enantiomers were described to be separately pharmacologically active, and the preferred non-racemic mixture is observed to have the expected additive properties. This differs substantially from the unexpected synergistic effects that we observed with dimiracetam. 

Many effects have been investigated to understand the molecular mechanisms of action of nootropic drugs [54]. For racetams, modulation of glutamate signaling was a recurring theme. For example, oxiracetam and aniracetam increase the NMDA-induced release of glutamate, noradrenaline, and acetylcholine in synaptosomal brain preparations, preventing the inhibition of NMDA-induced release by kynurenic acid [19,20,49]. Racetams also modulate AMPA signaling [55,56,57,58], reducing receptor desensitization and deactivation to enhance synaptic plasticity [59,60]. Racemic dimiracetam, on the other hand, counteracts the NMDA-induced release of [^3^H]D-Asp in rat spinal cord synaptosomes. We used Zn^2+^ and ifenprodil to probe the involvement in this preparation of NMDA receptors containing a proton-sensitive GluN1 subunit and GluN2A and GluN2B subunits. While Zn^2+^ abolishes completely the dimiracetam effect, ifenprodil only partially affects it, suggesting the existence in the spinal synaptosomal preparation of two variant NMDA receptors linked to [^3^H]D-Asp release: one containing both the GluN2A and GluN2B subunits, the other containing the GluN2A subunit only [3].

Against this background, it is most interesting to note the published X-ray crystallographic model of piracetam bound to the AMPA GluA2 and GluA3 subunit domains [61], which suggests a possible way forward to try to understand the molecular mechanism underpinning our findings. This study showed that piracetam can bind in multiple copies to the S1S2 domains of GluA3i, GluA2i, and GluA2o, where six piracetam-binding sites are located at the dimer interface of the receptor. The dimer forms upon activation by an agonist [62] and represents the active receptor conformation. Preventing or favoring dimer dissociation can either block or accelerate desensitization and affect deactivation. This structure is consistent with previous evidence that nootropics act as allosteric modulators at AMPA receptors [58,61]. Looking for a possible explanation of the unorthodox findings reported in the present paper, and ruling out an unlikely pharmacokinetic explanation, given the superimposable pharmacokinetics of the racemate and the two enantiomers (data not shown), we hypothesize that dimiracetam enantiomers can stereo-selectively bind similar piracetam-binding sites which may be present in the NMDA receptor as well. In this way, the two dimiracetam enantiomers would cooperate to engage their target optimally and, providing the correct racemic ratio as in MP-101, would optimize the interaction with NMDA and the consequent allosteric modulatory effect. Whether direct evidence for such a mechanism can be obtained, and whether such a feature applies to other chiral members of the nootropic racetam family [63,64,65,66], remains to be investigated.

## 5. Conclusions

MP-101, a non-racemic enantiomeric mixture containing three parts of R-dimiracetam and one part of S-dimiracetam, is pharmacologically more potent than racemic dimiracetam in several animal models of neuropathic pain cognition and depression.

Additional work is needed to define the precise molecular details underpinning this experimental evidence. The demonstrated efficacy of MP-101 in the oxaliplatin-induced peripheral neuropathic pain model, together with the preclinically and clinically established safety and tolerability profile of racemic dimiracetam strongly support the development of MP-101 for the management of symptoms of peripheral neuropathy in CIPN, an area with a high, unmet medical need. 

## 6. Patents

US Patent 10738054, 11 August 2020.

## Figures and Tables

**Figure 1 cells-11-04027-f001:**
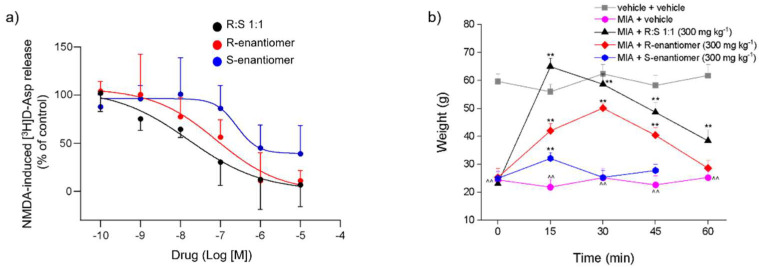
(**a**) Concentration-dependent (log[M]) effects of dimiracetam, R-dimiracetam, and S-dimiracetam on the release of [^3^H]D-Asp induced by NMDA in rat spinal cord synaptosomes in superfusion. Synaptosomes were labeled with the radioactive tracer, layered on microporous filters, and superfused as described in the Materials and Methods section. Results are expressed as percent of control. Results are expressed as percent of control conditions (NMDA-induced [^3^H]D-Asp release reported as 100% response). Data are means ± S.E.M. of 3–9 experiments run triplicate (three superfusion units for each experimental condition). (**b**) Evaluation of the effects of R- and S-enantiomers and R:S 1:1 against sodium monoiodoacetate (MIA)-induced mechanical hyperalgesia. MIA, 2 mg/25 µL, was injected into the tibio–tarsal joint of rats on day 1. Tested compounds were *per os* acutely administered 164 days after MIA administration at the dose of 300 mg kg^−1^. Control animals were treated with vehicles. Mechanical hyperalgesia was measured on the ipsilateral paw by the Paw pressure test. Results are expressed as mean ± S.E.M. of 20 rats analyzed in 4 different experimental sets. Statistical analysis was one-way ANOVA followed by Bonferroni’s post-hoc comparison. ^^ *p* < 0.0 vs. vehicle + vehicle-treated animals; ** *p* < 0.01 vs. MIA + vehicle-treated animals.

**Figure 2 cells-11-04027-f002:**
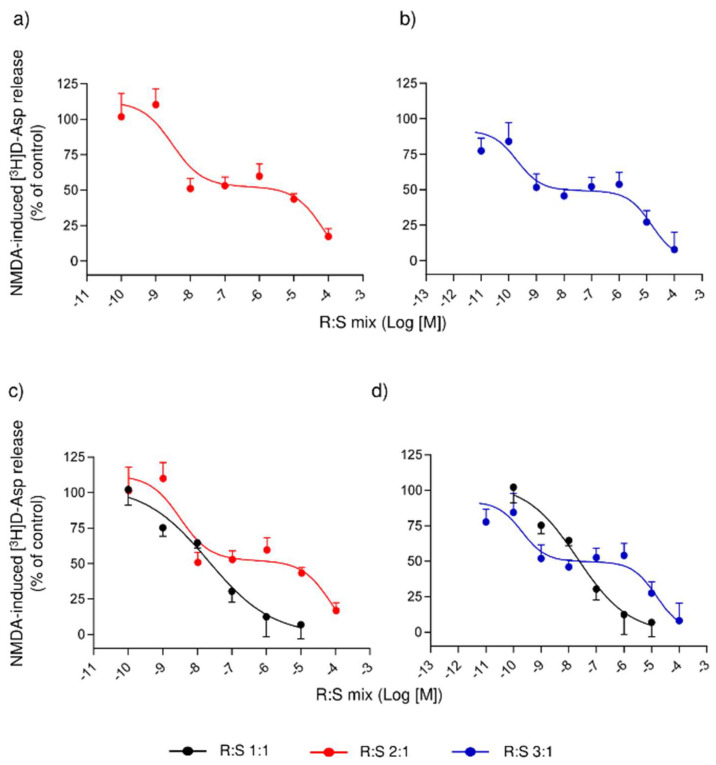
Concentration–response (log[M]) curves of non-racemic (**a**) 2:1 or (**b**) 3:1 R- over S-dimiracetam mixtures (R:S 2:1 and R:S 3:1) on the release of [^3^H]D-Asp induced by NMDA in rat spinal cord synaptosomes in superfusion. (**c**,**d**) Comparison of the R:S 2:1 and 3:1 mixture curves to dimiracetam. Synaptosomes were labeled with the radioactive tracer, layered on microporous filters, and superfused as described in the Materials and Methods section. Results are expressed as percent of control conditions (NMDA-induced [^3^H]D-Asp release reported as 100% response).. Data are means ± S.E.M. of 3–9 experiments run triplicate (three superfusion units for each experimental condition).

**Figure 3 cells-11-04027-f003:**
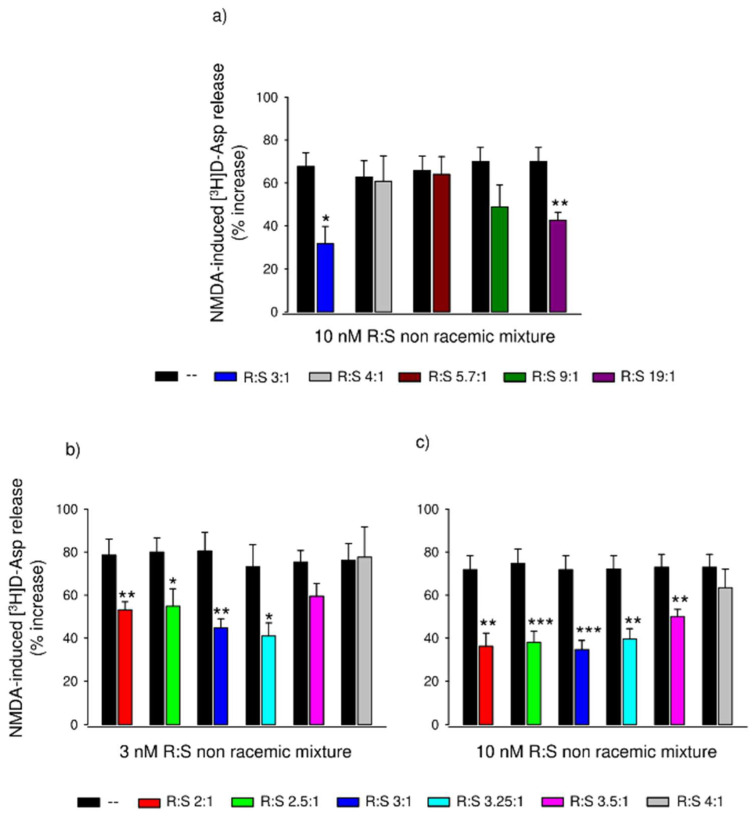
Effects of a series of non-racemic mixtures on the release of [^3^H]D-Asp induced by NMDA in rat spinal cord synaptosomes in superfusion. (**a**) effects of 10 nM non-racemic R/S mixtures ranging from 3:1 to 19:1; (**b**) effect of 3 nM non-racemic R/S mixtures ranging from 2:1 to 4:1 and (**c**) effect of 10 nM non-racemic R/S mixtures ranging from 2:1 to 4:1. Synaptosomes were labeled with the radioactive tracer, layered on microporous filters, and superfused as described under the Materials and Methods section. Results are expressed as percent increase in [^3^H]D-Asp release over basal efflux. Data are means ± S.E.M. of 3–9 experiments run triplicate (three superfusion units for each experimental condition). * *p* < 0.05, ** *p* < 0.01, *** *p* < 0.001, when compared to the respective NMDA control (two-tailed Student’s *t*-test).

**Figure 4 cells-11-04027-f004:**
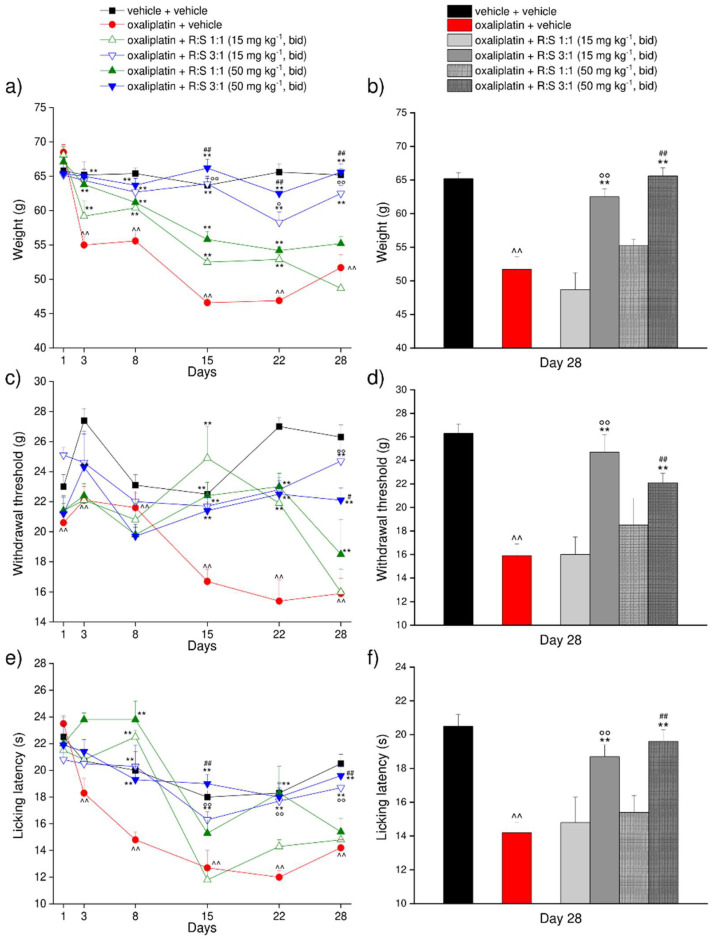
Evaluation of the effects of R:S 1:1 and R:S 3:1 against oxaliplatin-induced neuropathic pain. Oxaliplatin (2.4 mg kg^−1^ i.p.) was administered on days 1–15, for a total of 11 injections. R:S 1:1 and R:S 3:1 mixtures were dissolved in 1% CMC and were daily per os administered from day 1 until day 22 (15 and 50 mg kg^−1^ bid, in the morning and in the evening). Control animals were treated with vehicles. Behavioral measurements were performed before the daily injection of oxaliplatin or oral compounds. The pain-relieving effects of R:S 1:1 and R:S 3:1 was assessed by (**a**) Paw pressure test, (**c**) von Frey test and (**e**) Cold plate during treatment at the indicated time points, and (**b**,**d**,**f**) after 1 week of treatment wash-out (day 28, Paw pressure, von Frey and Cold plate tests, respectively). Each value represents the mean ± S.E.M. of 6 rats. Statistical analysis was one-way ANOVA followed by Bonferroni’s post-hoc comparison. ^^ *p* < 0.01 vs. vehicle + vehicle; ** *p* < 0.01 vs. oxaliplatin + vehicle; ° *p* < 0.05 and °° *p* < 0.01 vs. oxaliplatin + R:S 1:1 15 mg kg^−1^; ^#^ *p* < 0.05 and ^##^ *p* < 0.01 vs. oxaliplatin + R:S 1:1 50 mg kg^−1^-treated animals.

**Figure 5 cells-11-04027-f005:**
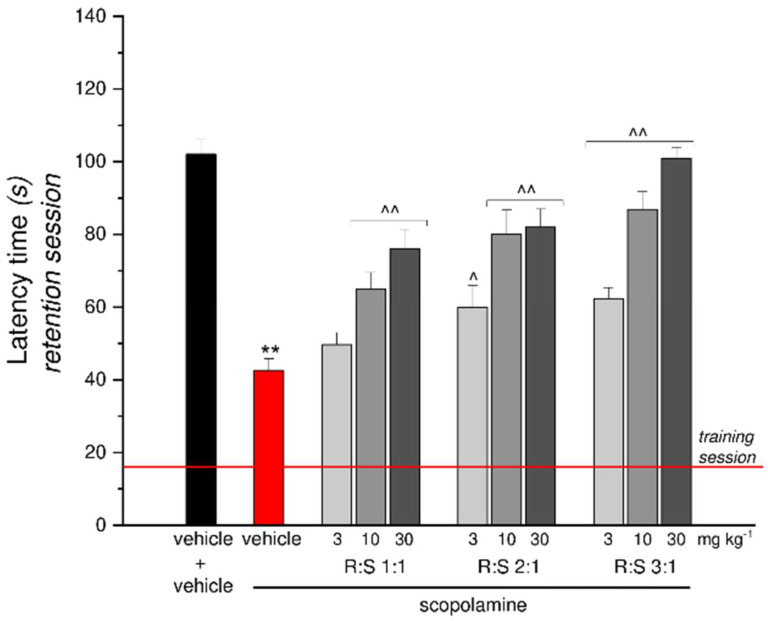
Effect of R:S 1:1, R:S 2:1 and R:S 3:1 against scopolamine-induced amnesia in mice in the Passive avoidance test. The latency (s) to enter the dark chamber during the retention test is reported. The red line represents the latency time for inexperienced mice recorded during the training session. Compounds R:S 1:1, R:S 2:1 and R:S 3:1 at different dosages (3 mg kg^−1^, 10 mg kg^−1^ and 30 mg kg^−1^) were *per os* administered 20 min before the training session while scopolamine (1.5 mg kg^−1^ s.c.) was injected immediately after. Data are expressed as mean ± S.E.M. of 12 mice analyzed in 2 different experimental sets. Statistical analysis was one-way ANOVA followed by Bonferroni’s post-hoc comparison. ** *p* < 0.01 vs. vehicle + vehicle; ^ *p* < 0.05 and ^^ *p* < 0.01 vs. scopolamine + vehicle.

**Figure 6 cells-11-04027-f006:**
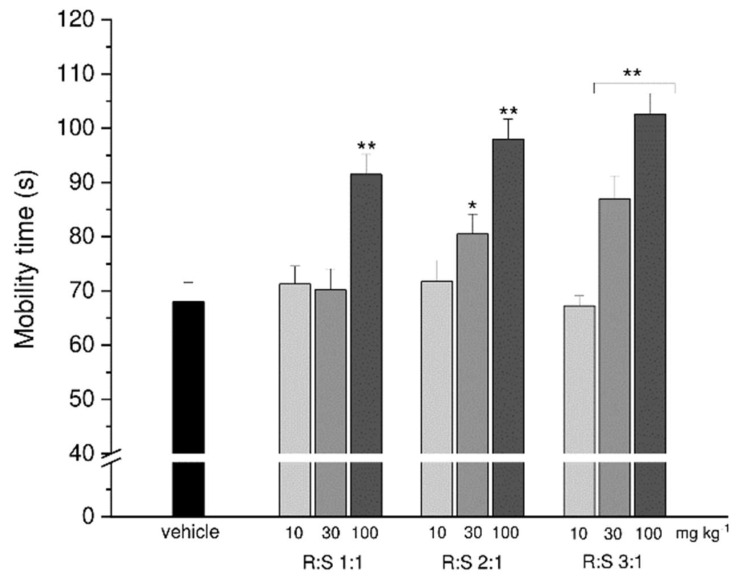
Effect of R:S 1:1, R:S 2:1 and R:S 3:1 on depressive-like behavior in mice in the forced swimming test. Mobility time (s) was recorded during the last 4 min of the test. Compounds R:S 1:1, R:S 2:1 and R:S 3:1 at different dosages (10 mg kg^−1^, 30 mg kg^−1^ and 100 mg kg^−1^) were administered *per os* 25 min before the beginning of the test. Data are expressed as mean ± S.E.M. of the results obtained from 12 mice analyzed in 2 different experimental sets. Statistical analysis was one-way ANOVA followed by Bonferroni’s post-hoc comparison. * *p* < 0.05 and ** *p* < 0.01 vs. vehicle + vehicle.

## Data Availability

The data presented in this study are available on request from the corresponding author upon reasonable request.

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
