# Peer review of "Pharmacological Profile of MP-101, a Novel Non-racemic Mixture of R- and S-dimiracetam with Increased Potency in Rat Models of Cognition, Depression and Neuropathic Pain"

_cells, 2022, doi:10.3390/cells11244027_

Round 1

Reviewer 1 Report

The work presented by Bonifacino et al reports interesting results on the efficacy of dimiracetam (mixture of enantiomers) on cognition, depression, and neuropathic pain. The principle aim seems to be focused on neuropathic pain (oxaliplatin-induced peripheral neuropathy), for which innovative therapies are more than necessary.

The results underlined the unexpected efficacy of the mixture (R:S 3:1) against the racemic one.

I globally suggest some explanations and corrections in the manuscript.

Here are my comments:

In the abstract, but also in the manuscript, it is unclear when the name dimiracetam is used, if it corresponded to the racemic or the mixture. Please explain.

Line 67, what is the “2ainite-gated ion channel”, Is it a typo?

Line 216 and 235 “6ehavior”, typo?

What is the rational for the selection of the various mixture of dimiracetam and doses?

For experiment in MIA model, what is the rational for the R:S 1:3 mixture, this one has not been tested before in in vitro assays.

For the in vivo assay with the MIA model, I was thinking that this model is not a neuropathic pain one. Can you explain this choice, please?

In comparison with the oxaliplatin model, doses of dimiracetam are not the same, and finally quite different 15 and 50 mg/kg bid vs 300 mg/kg. can you explain?

Moreover, the doses of 15 and 50 mg/kg bid are lower than in previous work (doi: 10.1016/j.neuropharm.2014.01.029; 10.1016/j.neuro.2015.08.002 ; 10.1007/s00280-017-3449-8) can you explain/justify, please ?

In the figure 4, if I understand well, the figures b) d) and f) are just replication of the last measure on day 28. It is probably unnecessary to repeat these data.

Not sure that the Figure S2, is necessary.

The results on mice models (scopolamine and force swimming) are interesting, however, it seems a little bit embarrassing to mix rat and mouse studies exploring different mechanisms and different ranges of doses (not sure that rat and mice behave in the same way for the same test).

For all the results, do the authors have information on possible sedative effect of dimiracetam (and also mixture) ?

Author Response

Cells

Special Issue “Frontiers in Neuroinflammation”

Florence, December 5th 2022

Dear Editor,

We wish to thank you and the reviewers for the careful review of our manuscript entitled “Pharmacological profile of MP-101, a novel non-racemic mixture of R- and S-dimiracetam with increased potency in rat models of cognition, depression and neuropathic pain” and for your decision to consider a modified version. Please find enclosed the revised manuscript (with changes marked) according to the reviewer’s suggestions.

Reviewer 1

The work presented by Bonifacino et al reports interesting results on the efficacy of dimiracetam (mixture of enantiomers) on cognition, depression, and neuropathic pain. The principal aim seems to be focused on neuropathic pain (oxaliplatin-induced peripheral neuropathy), for which innovative therapies are more than necessary.

The results underlined the unexpected efficacy of the mixture (R:S 3:1) against the racemic one.

We globally suggest some explanations and corrections to the manuscript. Please find below our comments:

Point 1. In the abstract, but also in the manuscript, it is unclear when the name dimiracetam is used, if it corresponded to the racemic or the mixture. Please explain.

Response. We apologize if the terminology used was unclear. The racemic nature of the term “dimiracetam” is clearly defined in the first line of the introduction. Throughout the document, we use the term “dimiracetam” to refer to the racemic mixture, i.e. 1:1 R-to-S; while with the term MP101 we refer to the 3:1 R-to-S non-racemic mixture. We have modified the manuscript to make this distinction clearer for the reader.

Point 2. Line 67, what is the “2ainite-gated ion channel”, Is it a typo?

Response. We apologize for the typographical error, the text has been corrected accordingly.

Point 3. Line 216 and 235 “6ehavior”, typo?

Response. Once again we apologize for the typographical errors; the text has been corrected accordingly

Point 4. What is the rationale for the selection of the various mixture of dimiracetam and doses?

Response. Our rationale for the various mixtures of dimiracetam enantiomers we studied in vitro derived simply from mathematical progression: enantiomeric ratios of 2:1, 3:1, and 4:1 R-to-S correspond to enantiomeric excesses of 33%, 50%, and 60% of the R enantiomer, respectively. When we further considered the effect of pure R and of S dimiracetam (each with >95% enantiomeric excess), it was clear that more closely spaced comparisons around the 3:1 R-to-S ratio would be interesting, because it would allow us to define the range of ratios whose potency would exceed that of racemic dimiracetam. 

The rationale for the doses of racemic dimiracetam and the 3:1 R-to-S ratio was based on our previous detailed studies with racemic dimiracetam. The dose response of racemic dimiracetam after a single oral administration has previously been published (Fariello et al, Neuropharmacology. 2014 Jun;81:85-94). With sustained twice-daily oral administration, the dose response relationship of racemic dimiracetam has also been reported: the plateau of the dose-response curve is approximately 150 mg/kg per os, while the ED50 is approximately 75 mg/kg. In the present study, we show that the R-enantiomer is more potent than the S-enantiomer, and that the ED50 of the R-enantiomer is approximately 150 mg/kg, while the ED50 of the S-enantiomer appears to lie above 300 mg/kg. It is noteworthy that the R-enantiomer and racemic dimiracetam are equi-efficacious when used at high doses: maximal efficacy can be achieved with high doses of either the R-enantiomer or of racemic dimiracetam. When comparing the 3:1 R-to-S mixture, we therefore deliberately chose a *low* dose (50 mg/kg)  or a *very low* dose (15 mg/kg) of either the 3:1-mixture or racemic dimiracetam, as at high doses the two would appear to be equi-efficacious.

Point 5. For the experiment in the MIA model, what is the rational for the R:S 1:3 mixture, this one has not been tested before in in vitro assays.

Response. We thank the referee for the opportunity to clarify this point. It is correct that we conducted most of the in vitro testing with mixtures consisting of an excess of the R-enantiomer; we have conducted fewer in vitro tests with an excess of the S-enantiomer. We note, however, that the there is a very good correspondence in the rank order of potency of the various mixtures between the in vitro and the in vivo assays. However, we considered it to be too ambitious to expect this matching between in vitro and in vivo results to extend to in vivo results with small differences in the composition of the administered mixtures. Therefore, we chose only one mixture with an excess of the S-enantiomer for in vivo testing, i.e. the 1:3 R-to-S mixture.

Point 6. For the in vivo assay with the MIA model, I was thinking that this model is not a neuropathic pain one. Can you explain this choice, please?

Response. We are grateful to the reviewer for the possibility to clarify this point. The MIA model is a preclinical model of articular pain that resembles human osteoarthritis (Ivanavicious et al., Pain. 2007;128:272–282, Pomonis et al., Pain. 2005;114:339–346). Previous studies using this model have demonstrated that intra-articular MIA results in decreased weight bearing on the injured limb, movement-evoked pain, and hypersensitivity to acute application of noxious (hyperalgesia) and non-noxious (allodynia) stimulation to the hindpaw, indicating referred pain (Combe et al., Neurosci Lett. 2004;370:236–240; Honore et al. Pain. 2009;142:27–35). In contrast to the Complete Freund’s Adjuvant-induced rheumatoid arthritis model, the MIA model results in the development of persistent pain that responds well to drugs like gabapentin and amitriptyline on days 14, 21 and 28 post-MIA but not to common anti-inflammatory drugs like naproxen and celecoxib (Ivanavicious et al., Pain. 2007;128:272–282). Collectively, these data lead us to the view that the MIA model is associated with an inflammatory state immediately after MIA injection, that is then followed by articular cartilage damage and subchondral bone lesions at days 8-14 onwards that resemble a neuropathic pain state, rather than an inflammatory one. We therefore carried our tests of dimiracetam and of the non-racemic mixtures late in the course of the MIA model, at a time where we consider a neuropathic situation exists.

Point 7. In comparison with the oxaliplatin model, doses of dimiracetam are not the same, and finally quite different 15 and 50 mg/kg bid vs 300 mg/kg. can you explain?

Moreover, the doses of 15 and 50 mg/kg bid are lower than in previous work (doi: 10.1016/j.neuropharm.2014.01.029; 10.1016/j.neuro.2015.08.002; 10.1007/s00280-017-3449-8) can you explain/justify, please?

Response. Please note our responses to Question 4. The doses of 50 mg/kg and 15 mg/kg per os were deliberately chosen as doses of racemic dimiracetam that we know from our previously published work are low or very low in this oxaliplatin-based model. This allows us to use the same mg/kg dose of the 3:1 R-to-S mixture, in order to compare the magnitude of the efficacious response between the two different enantiomeric mixtures. We considered this manner of dose selection to be a more efficient use of our resources and our experimental rats to allow comparison of the potency of these two mixtures, than if we had separately determined the ED50’s for racemic dimiracetam and for the 3:1 R-to-S mixture.

We also want to point the reviewers’ attention to the higher ED50 of racemic dimiracetam following single oral doses when compared to sustained twice-daily oral administration. Following single oral doses, the timepoint of maximal effect is 0.5-1.0 h of administration; within 2-3 h the drug effect is no longer observed; the lowest dose that achieves maximal efficacy (i.e. paw-withdrawal threshold not different from sham-treated animals) is 150 mg/kg; and the ED50 is approximately 75 mg/kg across several rodent models of neuropathic pain. With sustained twice daily oral administration, the duration of the maximum response increases progressively over 1-2 weeks, such that the maximal effect is sustained throughout the dosing interval. There are no signs of tachyphylaxis, but rather the opposite: the twice-daily oral dose of 75 mg/kg can be diminished to 50 mg/kg once daily, without signs of diminished efficacy out to 28 days of continuous dosing. Upon treatment cessation, the drug effect is evident for a period of up to 3 days, which significantly exceeds the expected disappearance of the dimiracetam enantiomers from the blood plasma compartment.

We believe the reviewers will also welcome a short summary of the pharmaceutic and pharmacokinetic properties of racemic dimiracetam. Racemic dimiracetam is highly water soluble and it is completely absorbed after oral administration as an aqueous solution. There is little to no metabolism of either enantiomer: they are excreted entirely by the kidney, and both have a terminal elimination half-life of 6-8 h. Their blood plasma pharmacokinetics exhibit a Cmax at 0.5-0.8 h. There is no in vivo interconversion of the enantiomers.

Point 8. In the figure 4, if I understand well, the figures b) d) and f) are just replication of the last measure on day 28. It is probably unnecessary to repeat these data.

Response. It is correct, the figures 4b, 4d and 4f are the replication of the measures reported on day 28 in the graphs on the left, however, it is important for us to stress these data since were collected after one week of wash-out. It is remarkable how the non-racemic mixture R:S 3:1 was still active at both dosages and how the effect was not only symptomatic but also preventive. We think that it can be easier for the reader to understand the effect of the treatments by keeping the figures 4b, 4d and 4f.

Point 9. Not sure that the Figure S2, is necessary.

Response. The Supplementary Figure S2 confirms the effects of R:S 1:1 and R:S 3:1 on mechanical hyperalgesia and mechanical and thermal allodynia 1 h after the morning administration at days 5, 8, 15 and 22 and endorses the superior potency of the R:S 3:1 over the R:S 1:1. So the chronic efficacy was demonstrated (24 h after the last administration), an effect that is not improved by an acute, novel treatment. We kindly request to keep this figure on the supplementary materials.

Point 10. The results on mice models (scopolamine and force swimming) are interesting, however, it seems a little bit embarrassing to mix rat and mouse studies exploring different mechanisms and different ranges of doses (not sure that rat and mice behave in the same way for the same test).

Response. By including our results from the scopolamine and the forced swimming tests, it was our deliberate intent to demonstrate that the preference for the 3:1 R-to-S mixture extends to higher brain structures. Whereas a drug effect in the spinal cord alone can explain the efficacy of racemic dimiracetam and of the 3:1 R-to-S mixture in models of neuropathic pain, there must also be an effect of racemic dimiracetam and of the 3:1 R-to-S mixture in higher brain structures, in order to achieve efficacy in these two CNS models. Furthermore, the rank order of potency is retained – even though the experiments are performed in mice, not in rats. We consider this to strengthen our interest to investigate potential utility of the 3:1 R-to-S mixtures in other CNS disorders. Here, too, we request to keep our results in the scopolamine and forced swimming models in the manuscript. Moreover we would ask to the referee to consider the presence of two animal species as a plus of this work.. In our experience, mice and rats in which we can reproduce the same neuropathic pain model (e.g. oxaliplatin) respond in the same way to a drug treatment with completely comparable results. However sometimes a preclinical model requires the use of one animal species over another and is the case of the mouse model of scopolamine-induced amnesia and the force swimming test.

Point 11. For all the results, do the authors have information on possible sedative effect of dimiracetam (and also mixture)?

Response. We are grateful to the reviewer for the possibility to clarify this point. Across all experiments we have performed we have seen neither neurological, motor alterations nor sedative effects evoked by dimiracetam or of the non-racemic mixture. Moreover, in our previous papers, we have already assessed the effect of dimiracetam on the animal’s motor skills and on several behavior and neurological parameters (Irvin test) without highlight any negative effect evoked by the treatment (Fariello et al., Neuropharmacology. 2014 81:85-94; Di Cesare Mannelli et al., Cancer Chemother Pharmacol. 2017 80(6):1091-1103).

Reviewer 2 Report

The paper by Bonifacino et al. show that MP-101 can reduce neuropathic pain as well as improve cognition and depression in animal models. This is a simple and nicely done studies. A few minor comments below:

1. The study lacks biochemical or molecular evidence as the authors correctly point out in the discussion. Protein (eg NMDA receptor analysis etc) or gene expression analyses after MP-101 administration would be helpful and would strengthen the study. 

2. In the model of neuropathic pain (MIA) it seems that the effects of MP-101 are rather temporary and short-lived. Can the authors comment on that? 

Author Response

Cells

Special Issue “Frontiers in Neuroinflammation”

Florence, December 5th 2022

Dear Editor,

We wish to thank you and the reviewers for the careful review of our manuscript entitled “Pharmacological profile of MP-101, a novel non-racemic mixture of R- and S-dimiracetam with increased potency in rat models of cognition, depression and neuropathic pain” and for your decision to consider a modified version. Please find enclosed the revised manuscript (with changes marked) according to the reviewer’s suggestions.

Reviewer 2

The paper by Bonifacino et al. show that MP-101 can reduce neuropathic pain as well as improve cognition and depression in animal models. This is a simple and nicely done studies. A few minor comments below:

Point 1. The study lacks biochemical or molecular evidence as the authors correctly point out in the discussion. Protein (eg NMDA receptor analysis etc) or gene expression analyses after MP-101 administration would be helpful and would strengthen the study. 

Response. We perfectly agree with the referee’s comment, the biomolecular analysis of 3:1 pharmacodynamic is very interesting. We reported here the peculiar higher effectiveness of the mixture R/S in several behavioral paradigms. This characteristic was maintained also in the negative modulation of NMDA-induced glutamate release in spinal synaptosomes. As suggested, NMDA receptor levels could be a relevant information even if our findings suggest a functional modification We have sought with little success in various settings biochemical or molecular evidence to support our observed modulation of glutamate signaling by dimiracetam and its 3:1 R-to-S mixture of enantiomers. In Xenopus oocytes and in mammalian cell lines engineered to overexpress various combinations of the heterotetrameric NMDA-gated and the homodimeric AMPA-gated ion channels, we have obtained no clear evidence for a modulatory effect of dimiracetam. In electrophysiological recordings in hippocampal rat brain slices we have obtained evidence of an inhibition of evoked NMDA-type neuronal discharges; but these results lack a molecularly-defined substrate or drug-target. We stand by our explanatory hypothesis for our results, which we have arrived at by analogy to the published x-ray crystallographic structure of multiple piracetam binding sites in the glutamate-binding domain of the AMPA-type ion channel: we believe our model of cooperative binding by glutamate-gated ion channels of several copies of R- and S-enantiomers in unequal numbers can provide a rational basis for understanding the observed preference for certain non-racemic mixtures of dimiracetam. Furthermore, gene expression could highlight novel pathway to explore; for this purpose a transcriptomic or proteomic analysis of different regions of the CNS could be the most indicated approach. Again, the efficacy of dimiracetam and its non-racemic mixture MP-101 after repeated treatments strongly suggests a disease-modifying profile through neuroprotective properties that should be evaluated by an extensive histopathological and molecular study. For all this reasons, we would like to dedicated a separate work on the clarification of 3:1 and dimiracetam pharmacodynamics and molecular profile since Our efforts to further explore these phenomena in settings with clearly defined molecular constituents will be performed, and we hope to be able to report more fully on these efforts in the future.

Point 2. In the model of neuropathic pain (MIA) it seems that the effects of MP-101 are rather temporary and short-lived. Can the authors comment on that? 

Response. Please see our response to Reviewer 1, under his / her Points 4 and 7, which we repeat in part here: “We…want to point the reviewers’ attention to the higher ED50 of racemic dimiracetam following single oral doses when compared to sustained twice-daily oral administration. Following single oral doses, the timepoint of maximal effect is 0.5-1.0 h of administration; within 2-3 h the drug effect is no longer observed; the lowest dose that achieves maximal efficacy (i.e. paw-withdrawal threshold not different from sham-treated animals) is 150 mg/kg; and the ED50 is approximately 75 mg/kg across several rodent models of neuropathic pain. With sustained twice daily oral administration, the duration of the maximum response increases progressively over 1-2 weeks, such that the maximal effect is sustained throughout the dosing interval. There are no signs of tachyphylaxis, but rather the opposite: the twice-daily oral dose of 75 mg/kg can be diminished to 50 mg/kg once daily, without signs of diminished efficacy out to 28 days of continuous dosing. Upon treatment cessation, the drug effect is evident for a period of up to 3 days, which significantly exceeds the expected disappearance of the dimiracetam enantiomers from the blood plasma compartment.”

“We believe the reviewers will also welcome a short summary of the pharmaceutic and pharmacokinetic properties of racemic dimiracetam. Racemic dimiracetam is highly water soluble and it is completely absorbed after oral administration as an aqueous solution. There is little to no metabolism of either enantiomer: they are excreted entirely by the kidney, and both have a terminal elimination half-life of 6-8 h. Their blood plasma pharmacokinetics exhibit a Cmax at 0.5-0.8 h. There is no in vivo interconversion of the enantiomers.”

Moreover, MP101is effective up to 45/60 min. This is a quite frequent duration of compound effects in rodents. Also drug long lasting in human show similar “temporary” profile in mice or rats due to higher metabolism of rodents. To note, in previous paper from this group, also reporting the activity of dimiracetam, clinically used drugs were tested showing a comparable duration (Fariello et al., Neuropharmacology. 2014 81:85-94; Di Cesare Mannelli et al., Neurotoxicology. 2015 50:101-7; Di Cesare Mannelli et al., Cancer Chemother Pharmacol. 2017 80(6):1091-1103).
